# Atypical Chronic Lymphocytic Leukemia—The Current Status

**DOI:** 10.3390/cancers15184427

**Published:** 2023-09-05

**Authors:** Tadeusz Robak, Anna Krawczyńska, Barbara Cebula-Obrzut, Marta Urbaniak, Elżbieta Iskierka-Jażdżewska, Paweł Robak

**Affiliations:** 1Department of Hematology, Medical University of Lodz, 90-647 Lodz, Poland; akrawczyn@onet.eu (A.K.); barbara_cebula@wp.pl (B.C.-O.); martau868@gmail.com (M.U.); elaiskierka@gmail.com (E.I.-J.); robakpawel@op.pl (P.R.); 2Department of General Hematology, Copernicus Memorial Hospital, 93-513 Lodz, Poland; 3Department of Hematooncology, Copernicus Memorial Hospital, 93-513 Lodz, Poland

**Keywords:** CLL, atypical CLL, CD5, CD23, CLL differential diagnosis

## Abstract

**Simple Summary:**

The degree of differentiation of leukemic cells seen in clinical practice is significant. The dissimilarities concern morphology, cytogenetics, and immunophenotype. This leads to a distinction between typical chronic lymphocytic leukemia and the atypical one, which is not taken into consideration in the standard diagnostic process yet. The heterogenic course of the disease encourages the search for prognostic factors that would aid in the selection of the most effective individual therapy for each patient. The atypical CLL constitutes a major difficulty in the diagnostic process and, therefore, in the choice of accurate treatment.

**Abstract:**

A diagnosis of typical chronic lymphocytic leukemia (CLL) requires the presence of ≥5000 clonal B-lymphocytes/μL, the coexistence of CD19, CD20, CD5, and CD23, the restriction of light chain immunoglobulin, and the lack of expression of antigens CD22 and CD79b. Atypical CLL (aCLL) can be distinguished from typical CLL morphologically and immunophenotypically. Morphologically atypical CLL cells have been defined mainly as large, atypical forms, prolymphocytes, or cleaved cells. However, current aCLL diagnostics rely more on immunophenotypic characteristics rather than atypical morphology. Immunophenotypically, atypical CLL differs from classic CLL in the lack of expression of one or fewer surface antigens, most commonly CD5 and CD23, and the patient does not meet the criteria for a diagnosis of any other B-cell lymphoid malignancy. Morphologically atypical CLL has more aggressive clinical behavior and worse prognosis than classic CLL. Patients with aCLL are more likely to display markers associated with poor prognosis, including trisomy 12, unmutated *IGVH*, and CD38 expression, compared with classic CLL. However, no standard or commonly accepted criteria exist for differentiating aCLL from classic CLL and the clinical significance of aCLL is still under debate. This review summarizes the current state of knowledge on the morphological, immunophenotypic, and genetic abnormalities of aCLL.

## 1. Introduction

Chronic lymphocytic leukemia (CLL) is characterized by monoclonal B cell proliferation as well as the accumulation of mature lymphocytes within various organs, including the peripheral blood (PB), bone marrow (BM), lymph nodes, and spleen [1,2]. It is the most common form of adult leukemia in the Western world, comprising 25 to 30 percent of all cases [3]. Its incidence has been estimated as 5.1 cases per 100,000 individuals [4], occurring about twice as often in men as in women. In the United States, an estimated 20,380 new cases (12,130 men and 6610 women) and 4490 deaths were reported for 2023 [5]. Most CLL patients are older, with a median age of 72 years at diagnosis [2] Only nine percent of newly diagnosed patients with CLL are younger than 45 years [6]. 

In most patients, CLL is diagnosed incidentally on a routine blood analysis. However, in some patients, lymphadenopathy, splenomegaly, anemia, or thrombocytopenia are observed. A diagnosis of CLL requires the presence of ≥5000 clonal B-lymphocytes/μL in the PB, together with CD5, CD19, and CD23 antigen expression and light-chain surface immunoglobulin (SIg) restriction, confirmed by flow cytometry, for a duration of at least three months [1,2]. 

In the recent International Consensus Classification (ICC) and the 5th edition of the World Health Organization Classification (WHO HAEM5), the diagnostic criteria for CLL/small lymphocytic lymphoma (SLL) are the same and include the coexistence of CD19, CD20, CD5, and CD23 antigens on the leukemic cells [7,8]. In some cases, CD10, CD43, CD49d, CD79b, CD81, CD200, and ROR1 can be useful for differentiating CLL from other lymphoproliferative disorders (LPD) [9]. The clonality of the circulating B-lymphocytes needs to be confirmed by flow cytometry. The WHO-HAEM5 classification of CLL includes prolymphocytic progression [8]. Moreover, it is important to distinguish accelerated CLL from diffuse large B-cell transformation (Richter syndrome) [10].

In 1989, the French–American–British (FAB) Cooperative Group proposed subdividing CLL into typical and atypical subtypes [11]. However, the clinical significance of atypical CLL (aCLL) is still debated. In 1999, Criel et al. proposed the concept of aCLL based on its unique morphology, immunophenotype, genetic abnormalities, clinical features, and prognostic factors [12]. Since then, several studies have attempted to identify the causes of the poor prognosis in this group, such as abnormal lymphocyte morphology and karyotype (e.g., trisomy 12 and deletion 17p) [13]. However, atypical CLL is not included in the WHO HAEM5 nor the ICC classification [7,8]. Recent progress in elucidating the genetic characteristics of LPD has allowed for a clearer understanding and more accurate definition of CLL and its related disorders [14,15,16]. Therefore, the present review summarizes the current state of knowledge regarding the morphology, immunophenotype, and genetics of aCLL.

## 2. Atypical Morphological Feature

In blood smears from classic CLL, leukemic lymphocytes are characteristically small, mature lymphocytes with a narrow border of cytoplasm and a dense nucleus lacking discernible nucleoli and having partially aggregated chromatin (Figure 1A,B). However, there are currently no standard, commonly accepted criteria for the morphological differentiation between aCLL and classic CLL [17,18,19,20,21]. Morphologically atypical CLL cells were defined mainly as large, atypical forms (Figure 1C,D), prolymphocytes (Figure 2A,B), or cleaved cells (Figure 2C,D). Prolymphocytes, or larger atypical cells up to 55% of the blood lymphocytes, do not exclude a CLL diagnosis [22]. In some patients with atypical CLL, larger cells and/or cells with abundant cytoplasm are present. This subtype of CLL is commonly associated with the trisomy of chromosome 12 and a higher intensity of sIg expression [12,13,22].

The FAB Cooperative Group defined three cytology-based subtypes of CLL. These comprise one classical form of CLL and two mixed-cell types: one CLL/prolymphocytic leukemia (PLL) form with 10–55% prolymphocytes in the PB smear and another form with larger cells characterized by lymphoplasmacytic features and more abundant cytoplasm-cleaved cells. The latter was subsequently called aCLL [11,12]. In the FAB classification, patients with ≥55% prolymphocytes are diagnosed with B-cell prolymphocytic leukemia (B-PLL). Cases with 15–55% of the prolymphocytes previously called CLL/PLL are now known as aCLL [21,23]. However, before the era of multiparameter flow cytometry, aCLL probably included patients with other LPDs with similar morphological features, such as mantle cell lymphoma (MCL) or follicular lymphoma (FL). Criel et al. included cases with large lymphocytes, prolymphocytoid cells, and cells with nuclear clefts in morphologically atypical CLL [12]. Large cells in aCLL have a similar morphology to classic CLL cells, but they are typically at least twice the size of erythrocytes and are morphologically similar to typical CLL cells (Figure 1). Melo et al. defined prolymphocytoid cells as being more pleomorphic and more heterogenous in size, with an irregular shape and large nucleolus compared to prolymphocytes in PLL [24]. Two types of CLL cells with nuclear clefts were identified: the first are small, while the second are twice as large as erythrocytes [25,26,27,28]. Several studies have indicated clinical, immunophenotypical, and genetic differences between cytological aCLL and typical CLL [18,29,30]. Frater et al. compared the morphologic, clinical, and immunophenotypic characteristics between patients with atypical and typical morphologic features of CLL [18]. Morphologically atypical cells were defined as deeply clefted lymphocytes larger than typical CLL with condensed chromatin and without prominent nucleoli.

A minimum of 10% of lymphocytes with clefted and folded nuclei in the PB were required for a diagnosis of aCLL. However, patients with circulating prolymphocytes or with increased numbers of prolymphocytoid cells were not included. Patients with a cytologically atypical CLL had a higher WBC count, a lower platelet count, and a significantly higher risk of disease progression. In addition, aCLL patients demonstrated a higher expression of CD23, but no significant difference was noted between aCLL and classic CLL regarding the expression of surface immunoglobulin (sIg), CD79b, or CD5. In addition, 11 of the patients with aCLL had trisomy 12. Morphologically, aCLL had more aggressive clinical behavior and worse prognosis than typical CLL [12].

In other studies, however, a strong correlation was found between atypical morphology, trisomy 12, and an aberrant immunophenotype [31]. During a study of the validity and clinical impact of CLL on a group of 390 patients, Criel et al. distinguished two closely related but different entities with different prognostic parameters and different survival rates [31]. Typical CLL cases were mostly diagnosed in Binet stage A/Rai 0; these patients did not need treatment at diagnosis and expected a longer survival. In contrast, aCLL cases were mainly diagnosed at a higher risk stage (Binet B/Rai I-II), mostly required immediate treatment, and were characterized by shorter survival. The most common chromosomal abnormalities were del(11q) (21% of those with an abnormal karyotype) in typical CLL and trisomy 12 (65%) in aCLL. The authors concluded that these two types of CLL can be identified based on a specific clinical presentation and different cytogenetic abnormalities and prognostic parameters. 

Finn et al. investigated the association between the morphology, immunophenotype, and karyotype of PB CLL lymphocytes in 26 patients with CLL [32]. Among the patients, seven of the eight (88%) with trisomy 12 had mixed cell morphology according to the FAB guidelines, compared to only three of the 18 (17%) without trisomy 12 (*p* = 0.004). In contrast, only one patient (12%) with trisomy 12 had typical CLL morphology. The atypical immunophenotype comprising strong CD20 expression, strong surface light chain expression, or the absence of CD23 expression was noted in six of the eight (75%) patients with trisomy 12, compared to only two of the 18 patients (11%) without trisomy 12 (*p* = 0.005). 

More recently, Marionneaux et al. used digital microscopy to classify CLL patients as morphologically aCLL or typical CLL on the archived blood films of 97 CLL patients [20]. In this study, the CLL/PLL and mixed-type subgroups were classified as aCLL. Atypical CLL was identified in 26 of 97 (27%) CLL cases, including 11 patients (42.3%) with CLL/PLL and 15 patients (57.7%) with the mixed-cell type. All patients with aCLL had at least one prognostic negative cytogenetic abnormality associated with a poor prognosis, compared with 34% of typical CLL patients. The patients with aCLL were more likely to have trisomy 12, unmutated *IGVH*, and CD38 expression compared with typical CLL. 

A number of studies have found morphoclinically aCLL to demonstrate more aggressive clinical behavior and a worse prognosis than classic CLL [23,27,33,34,35]. Ahn et al. evaluated the clinical and prognostic significance of 121 patients with CLL, 9 patients with aCLL, and 11 patients with B-PLL [28]. They found that lymphadenopathy was more common in CLL (42%) and aCLL (56%) than in B-PLL (0%) patients. In contrast, splenomegaly was more common in B-PLL patients (100%) than in CLL (25%) or aCLL (33%) patients. In addition, aCLL patients demonstrated more severe anemia, elevated lactate dehydrogenase, and β2-microglobulin than those with CLL or B-PLL. Importantly, patients with B-PLL commonly displayed CD5 or CD23 negativity, FMC7 positivity, and strong CD22 positivity. Patients with aCLL showed higher frequencies of FMC7 expression and a stronger expression of CD22 than those with classic CLL. In addition, the patients with aCLL or B-PLL had a worse prognosis than those with classic CLL. The overall survival (OS) rates after 10 years were 22.2% in aCLL, 46.3% in B-PLL, and 65.6% in classic CLL, respectively. These studies suggest that classic CLL and aCLL represent two closely related but different entities.

## 3. Atypical Immunophenotype

Currently, CLL is usually identified based on immunophenotypic characteristics. The typical immunophenotype includes the coexistence of CD19, CD20, CD5, and CD23 antigens on leukemic cells (Figure 3). While CLL can be accurately diagnosed using flow cytometry in the majority of patients, such a diagnosis becomes more challenging when CD23 or CD5 is not expressed by the leukemic cells [18,29,35,36,37,38,39]. In clinical practice, in cases lacking CD5 or CD23, the diagnostic process begins with confirming or rejecting aCLL.

### 3.1. CD5-Negative CLL

CD5-negative (Figure 4) and CD23-negative (Figure 5) variants of CLL are most commonly considered as aCLL. CD5 is expressed on the subset of normal T lymphocytes and in a subset of B-cells [40].

While most CLL patients demonstrate an expression of CD5 on the surface of neoplastic cells, this is absent in 7 to 20% of cases [41]. The diagnostic criteria for CD5-negative CLL is an important issue. Several studies have established a diagnosis of CD5-negative CLL where fewer than 5% of leukemic cells demonstrated CD5 expression [41,42,43,44,45]. CD5-negative B-CLL seems to have a different clinical presentation to CD5-positive CLL (Table 1) [41,42,43,44,45,46]. Based on a study on 423 CLL patients, Friedman et al. propose that the mean fluorescence intensity of the CD5 antigen on CLL cells may correlate with the clinical course of the disease: high mean fluorescence intensity rate correlated with longer progression-free survival (PFS) [44]. The CD5-negative patients frequently expressed a higher level of surface immunoglobulin and were more likely to present with splenomegaly [41].

The largest study, reported by Cartron et al. [41], compared 42 consecutive patients with CD5− CLL, observed from 1985 to 1991, with 79 with CD5+ CLL. However, this study was performed before the immunochemotherapy era, and the patients were treated either with chlorambucil or with CHOP (cyclophosphamide, doxorubicine, vincristine, and prednisone). At the time of the data analysis, the median survival time was not reached and no significant difference was observed between groups: the overall survival was 52% after 120 months in the CD5- group, and 66% after 90 months in the CD5+ group (*p* = 0.97). 

In another study, CD5-negative patients were found to exhibit milder disease symptoms and longer survival than the CD5-positive patients [42]. Efstathiou et al. compared the clinical and biological characteristics of 29 CD5-negative CLL patients with a control group of 29 sex- and age-matched, consecutive CD5-positive CLL patients [42]. In the CD5-negative group, splenomegaly, lymph node involvement, and hemolytic anemia were significantly less common than in the CD5-positive group. In this study, CD5− patients had a more favorable prognosis than the CD5+ CLL patients [42]. The patients in both groups were treated with chlorambucil and prednisone, fludarabine, or COP (cyclophosphamide, vincristine, and prednisone). The CD5- patients had a significantly longer median OS (97.2 months) than the CD5+ patients (84.0 months, *p* = 0.0025). 

In a more recent study, Demir et al. compared 19 consecutive CD5-negative CLL cases observed from 2009 to 2015 with 105 CD5-positive CLL patients [43]. Lymphadenopathy was less common (31.5%) in the CD5-negative group than the CD5-positive group (51.4%; *p* = 0.029), but splenomegaly was more common in the CD5-negative group (42.1%) than the CD5-positive group (16.1%, *p* = 0.029). However, no difference in Binet staging or the median neutrophil count was noted between the groups. The CD5-negative patients also exhibited a higher mean lymphocyte count (43.2 × 10^9^/L) than the CD5-positive patients (36.7 × 10^9^/L, *p* = 0.001). The five-year survival rate was 84.2% in the CD5-negative patients and 90.5% in the CD5-positive patients (*p* = 0.393). However, no treatment details are available. 

Romano et al. analyzed a cohort of 400 CLL patients, including 13 with a CD5-negative phenotype. No significant differences in the clinical course and survival were observed between the CD5-positive and CD5-negative CLL cells [46]. Furthermore, Kurec et al. found that CD5-negative CLL patients exhibited a lower hemoglobin level and higher disease stage (Rai’s classification) at diagnosis; they also demonstrated worse OS, with a five-year survival rate of 55% in CD5-negative patients and >90% in CD5-positive patients [45]. 

### 3.2. CD23-Negative CLL

The CD23 surface glycoprotein is another important antigen in CLL diagnosis. It is involved in the activation and proliferation of normal B lymphocytes and plays a role in CLL pathogenesis [47]. In particular, elevated CD23 isotype expression appears to have a protective effect on neoplastic B-cells, with the CD23 isotype stimulating them [48]. In addition, while a diagnosis of CLL requires the co-expression of CD23 with CD5 and CD19, MCL is based on the co-expression of CD5 and CD19 alone, without CD23 [49,50]. However, the atypical forms of both diseases have been recorded, with CD23-negative CLL (Figure 5) and CD23-positive MCL being reported [51]. Barna et al. investigated the cut-off levels of CD23 positivity and intensity in the differential diagnosis of CLL (84 patients) and MCL (26 patients) using flow cytometry analysis [51]. It was found that higher CD23 positivity (>92.5%) and/or mean fluorescence intensity (MFI) of CD23 greater than 44.5 correlated with a diagnosis of CLL. In contrast, CD23 positivity < 30% indicated a diagnosis of MCL. However, patients with CD23 positivity between 30 and 92.5% and intensity below 44.5 MFI can be seen both in CLL and MCL. In such cases, it is necessary to perform FISH for the translocation t(11;14) or immunohistochemical detection of cyclin D1 overexpression to differentiate CLL from MCL [51].

Some investigators indicate that the levels of CD23 may influence the prognosis of CLL patients [52,53]. Jursic et al. compared the level of expression of CD23 antigen and the clinical course of the disease in 77 previously untreated patients with CLL [47]. The results indicated a correlation between a lower level of CD23 expression and the number of lymphocytes in PB, and patients with >40% CD23 expression demonstrated longer progression-free survival (PFS) and OS. Further studies are needed, probably with genetic analysis, to unequivocally characterize CD5-negative and CD23-negative aCLL. 

## 4. Atypical Genotype

As mentioned above, morphologically atypical CLL is often associated with genetic abnormalities, most frequently with trisomy 12 [23,31,52,53]. Some studies indicate a strong association between trisomy 12 and atypical morphology in CLL [52]. However, this abnormality is not limited to lymphocytes with atypical morphology but can also occur in typical CLL cells. In addition, several mutations are known to render the leukemia more aggressive, such as TP53, ATM, SF3B1, FBXW7, POT1, CHD2, RPS15, IKZF3, ZNF292, ZMYM3, ARID1A, and PTPN11.7 mutations [22,54,55]. In total, 44 recurrently mutated genes and 11 recurrent somatic copy number variations have been recognized so far, including NOTCH1 and MYD88. The (11;14)(q13;q32) translocation was previously considered to be the hallmark of MCL [56,57]. However, it can also be present in 2–5% of patients with CLL. These patients have a poor prognosis due to an oncogenic IGH/CCND1 translocation and aberrant expression of cyclin D1 [56]. Trisomy 12 is one of the most frequent chromosomal abnormalities in CLL. Patents with CLL and t(11;14) commonly have an atypical morphology consisting of several small lymphocytes and some larger lymphocytes and prolymphocytes and share some biological characteristics with MCL [58,59]. In addition, these patients have an atypical immunophenotype comprising CD5+, CD19+, CD23+, SIg+, FMC7+, and CD10−. FMC7 expression is not present in classic CLL but has been observed in MCL. Moreover, surface Ig has variable expression in classic CLL. 

Atypical CLL with t(11;14) has a poorer prognosis than other CLL subtypes [52,59], and as these patients require prompt treatment, a fast and accurate diagnosis is needed. However, the optimal management of aCLL with t(11;14) is not established yet due to its low incidence [60,61]. 

Trisomy 12 was most frequently found in morphologically atypical CLL. One study examined trisomy 12 clones in CLL lymphocytes with atypical morphology based on MGG/FISH with standard cytomorphology [52]. Peripheral blood specimens were studied in four patients with aCLL using a DNA probe against the pericentromeric region of chromosome 12. Although trisomy 12 was identified in 10–34% of the lymphocytes, the disorder is not confined to lymphocytes with atypical morphology; it is also observed in typical CLL cells.

## 5. Scoring Systems for CLL Diagnosis

While several scoring systems have been developed for CLL diagnosis, Matutes et al. developed one on the basis of the most common CLL marker profile [61]. This system includes five markers known to be important for CLL diagnosis: CD5 positive, CD22 weak or negative, CD23 positive, FMC7 negative and Slg weak. For each of these five markers, a value of 1 or 0 is added according to whether it is typical or atypical for CLL, and the total score ranges from 0 in atypical CLL to 5 in typical CLL. Among the 400 analyzed cases, 87% scored 5 or 4, and only 0.4% scored 0 or 1. In other B-cell leukemias, including prolymphocytic leukemia, hairy cell leukemia (HCL), and HCL variants, 89% of the patients scored 0 or 1. Moreover, higher scores were found in CLL cases with more typical morphology (*p* < 0.0015). Using this panel of five standard markers, the accuracy of the scoring system for distinguishing CLL from non-CLL LPD was 91.8%, using a cutoff of four points or higher. 

Subsequently, Moreau et al. investigated whether the Matutes scoring system could be improved with the monoclonal antibody SN8 (CD79b) [62]. Briefly, PB samples were taken from 298 patients with CLL and 166 patients with non-CLL and analyzed using the five standard markers (CD5, CD22, CD23, FMC7, and Slg), together with the SN8 monoclonal antibody, using flow cytometry. SN8 recognizes an extracellular epitope on the B29 protein of the B-cell antigen receptor and has been clustered as CD79b. The addition of SN8 and a cut off of 4 points or higher, as in the original Matutes scoring system, increased the accuracy of the scoring system from 91.8% to 96.6%. However, a similar accuracy (96.8%) was observed if CD22 was excluded and a cutoff of 3 points or higher was used. Thus, the replacement of CD22 by an SN8 antibody in the original Matutes scoring system significantly increases the possibility of a discrimination between CLL and other B-cell LPD. Some patients with CLL show an immunophenotype that overlaps with other LPDs, especially those scoring 3 of 5 points in the Moreau system [62]. However, these classifications cannot define aCLL [17]. 

In other studies, the name aCLL has been used for CLL cases with uncommon laboratory features. Sandes et al. characterized patients with aCLL as having a Moreau score of 3 or less or lacking CD23 expression [37]. However, even now, the definition of aCLL remains controversial. 

The most recent immunophenotypic European Research Initiative on the CLL (ERIC) criteria for a diagnosis of CLL does not include any definition of aCLL [9]. The ERIC and European Society for Clinical Cell Analysis (ESCCA) Harmonization project study examined the diagnostic criteria for CLL based on its morphology and immunophenotype. The authors classified 14 of 35 potential markers as “required” or “recommended” for CLL diagnosis. It was agreed that the following diagnostic markers were required: CD19, CD5, CD20, CD23, SIg Kappa, and Lambda. These markers are consistent with the current diagnostic criteria and are used in clinical practice. The markers CD43, CD79b, CD81, CD200, CD10, and ROR1 appear to be potentially useful for differential diagnosis and are recommended as such. Immunophenotypically, atypical CLL correlates with higher Sig expression, FMC7 positivity and, typically, a lower Matutes score [31,53,61]. 

A new scoring system for CLL diagnosis in Chinese patients was developed recently by Li et al. [63]. In this system, CD5 and CD23 antigens were replaced with CD43 and CD180. Samples from 237 patients with diagnosis of mature B-cell LPD were randomly included into an exploratory and a validation group. The expression of CD5, CD19, CD20, CD23, CD43, CD79b, CD180, CD200, FMC7, and SIg were analyzed in all the samples. A sensitivity of 91.8% and specificity of 83.1% were calculated based on the receiver operating characteristic curves or ROC curves. These results were confirmed in a validation cohort (with a sensitivity of 90.5% (*p* = 0.808) and a specificity of 79.5% (*p* = 0.639)). This new CLL score improved the sensitivity to 79.4% in the CD5-negative or CD23-negative CLL group compared to the Moreau score (41.2%) and CLL flow score (47.1%). In the aCLL group, the new CLL improved the sensitivity to 84.2% compared to the Moreau score (61.4%) and CLL flow score (64.9%). This proposed atypical CLL score helped to offer an accurate differentiation of CLL from non-CLL, together with morphological and molecular methods, particularly in Chinese patients with an atypical immunophenotype [63]. 

In recent years, immunophenotype and genetic/molecular analysis have gained significance in the diagnosis of CLL at the expense of morphology, and this trend is likely to continue; indeed, recent guidelines tend to recommend an assessment of immunophenotype in the diagnosis of CLL rather than morphology. A significant diagnostic challenge is presented by CLL lacking CD5 or CD23 antigen expression, defined as aCLL; such cases need further laboratory and clinical investigation. In addition, cytogenetic and molecular data, especially concerning *TP53* and *IGHV* mutational status, can influence treatment decisions, even in the era of targeted drugs. The identification of less common genetic abnormalities should allow for a more accurate representation of aCLL and influence treatment selection in the future.

## 6. Differential Diagnosis of Atypical CLL with Other Lymphoproliferative Disorders

In patients with aCLL, additional markers should be used for differentiating CLL from other chronic B-cell neoplasms, especially MCL [64]. Classic MCL cases are characterized by a homogeneous population of small- to medium-sized lymphocytes with irregular nuclear contours [65,66]. However, both CLL and MCL are characterized by the presence of CD5-positive neoplastic cells. In such overlapping cases, an accurate diagnosis can be facilitated by including a cytogenetic study for t(11;14), which is characteristic for MCL. In MCL, t(11;14), (q13;q32), and cyclin D1 overexpression is observed in >95% of cases. However, in some patients, the differential diagnosis of CLL/SLL from other CD5-positive small B-cell lymphomas can be challenging due to the presence of overlapping morphologic and immunophenotypic features [65]. Classic MCL cells are positive for cyclin D1 and SOX11 and negative for CD23 and CD200. However, some MCL patients express CD23 and CD200 but lack SOX11. These patients morphologically and immunophenotypically resemble CLL, and this can be referred to as CLL-like MCL. Qiu et al. compared the clinic-pathological features of 14 cases with CLL-like MCL and 33 classic CLL cases [66]. Patients with CLL-like MCL were found to have lower numbers of neoplastic cells in the PB and lower BM involvement. Moreover, more patients with CLL-like MCL had mutated IGHV.

In the majority of patients, CLL can be differentiated from MCL and other lymphoid malignancies using flow cytometry (Table 2) [67]. In particular, patients with CLL-like MCL are more likely to display moderate to high expression of B-cell antigens and the Sig light chain. In addition, while CLL cells mostly present dim to negative expression of B-cell antigens, such as CD20, CD22, and CD79b, dim expression is uncommon in CLL-like MCL. In addition, MCL cases are positive for cyclin D1 and SOX11 and negative for CD10, CD23, CD200, and LEF1 [67]. However, differential diagnosis may be challenging if CD23 is not expressed by the CLL lymphocytes or if CD23 is expressed by MCL, as the two entities share many similarities [68]. In such cases, MCL diagnosis should be confirmed by immunohistochemical cyclin D1 detection, together with other cytofluorimetric readings, as well as cytogenetic or molecular testing.

One particularly useful marker for differentiating between aCLL and MCL is CD200 [63,69]. It is a membrane glycoprotein belonging to the immunoglobulin superfamily expressed on a subset of T and B lymphocytes. CD200 is expressed on myeloma, plasma cells, and in most patients with CLL. However, in MCL it is totally absent or pressed by a small minority of CD5-positive cells [35,67,68,68,69]. In a recent study on patients with aCLL, Ting et al. investigated the expression of CD200 on mature B cell neoplasms using an eight-color flow cytometry in combination with a conventional panel of flow cytometry markers [35]. The study included 63 samples with CLL or an atypical CLL phenotype, 6 samples of MCL, and 40 samples of other mature B cell neoplasms. CD200 was expressed in all the CLL samples, whereas the MCL samples were dim or negative for CD200. CD200 is closely related to the expression of CD23. Among the seven aCLL cases identified using conventional flow cytometry, with Matutes scores ≤3, all were strongly positive for CD200 and negative for t(11;14) translocation. Mehrpouri et al. attempted to distinguish between patients with various lymphoproliferative disorders, including 91 CLL, 15 atypical CLL, 14 MCL, and 11 CD5-/CD10-lymphoma, using a panel of specific markers and flow cytometry [70]. They found that CD22, CD23, FMC-7, and CD5 expression could be used to diagnose CLL, MCL, and CD5−/CD10− lymphoma but could not differentiate MCL from atypical CLL. However, the expression patterns of CD38 and immunoglobulin light chain were found to differ between aCLL and MCL: CD38 was expressed in 92.8% of MCL and 1.1% aCLL patients, and lambda light chain was expressed in 85% of MCL and 0% aCLL. Moreover, all of the patients with aCLL expressed kappa light chain. In other study, Ho et al. attempted to distinguish between MCL and phenotypically atypical CLL patients presenting with typical MCL phenotypes; the method was based on a flow cytometric analysis and a FISH panel for the detection of t(11;14) and other cytogenetic abnormalities characteristic for CLL [65]. Another marker used to differentiate the typical CLL from other B cell chronic lymphoproliferative disorders, including aCLL, is CD45. In a study of the expression of CD45 in typical CLL, aCLL, CLL/PLL, HCL, B-PLL, and B cell-non-Hodgkin lymphoma (B-NHL), Maljaei et al. found lower CD45 density in typical CLL than in the other conditions, including aCLL [71]. The diagnostic value of CD45 in CLL has been confirmed by other authors [72].

Another antigen that may be useful in the differentiation of CLL from other lymphoproliferative disorders is CD43 [73,74,75]. CD43 is a transmembrane sialogclyoprotein expressed on a subset of B-cells, T-cells, monocytes, and granulocytes but not on follicular lymphoma or MCL. It is expressed in both aCLL and typical CLL [76]. Moreover, the positive rate of CD43 was higher in CLL than in MCL. A recent study showed that CD43 and CD200 can differentiate CLL from non-CLL LPD with higher accuracy than the Matutes scoring system [63,72]. In addition, no difference in CD43 and CD200 expression has been observed between classic CLL and aCLL patients. In another study, CD81, CD5, CD23, and CD200 were found to be useful markers for distinguishing CLL from other LPDs [77].

Lymphoid enhancer-binding factor (LEF1) staining was identified in up to 95% of patients with CLL [78,79]. Lower LEF1 expression was found in CLL patients with atypical immunophenotypic or morphological findings compared to classic CLL [78]. In contrast, LEF1 expression was only observed in 4 to 9% of MCL patients [78,79].

CD180 is a toll-like receptor homolog protein expressed on B cells [80]. Although several studies have confirmed its expression on indolent lymphoid malignancies, CD180 is weakly expressed in CLL and MCL compared with normal B cells [81,82]. However, CD180 has been suggested as a marker for aCLL [63]. The presence of double-positive CD43 and CD180, together with other B-cell surface markers, can improve the sensitivity of the diagnosis in CLL patients negative for CD5 or CD23 staining [63]. Despite this, it can be difficult to make a definitive diagnosis of aCLL based on laboratory and clinical data alone. Other markers for the differential diagnosis of CLL are CD79b and FMC7. A recent study found that CD79b and FMC7 tend to express negatively in CLL and positively in MCL [63]. 

Finally, SOX11 and t(11;14) are important markers for differentiating CLL from MCL [83,84,85], with SOX11 being consistently detected in MCL but not in CLL [85]. However, no data are available on the SOX11 expression in aCLL. 

## 7. Conclusions

Atypical CLL (aCLL) is characterized by morphologic, phenotypic, and cytogenetic differences compared to classic CLL. However, aCLL is not formally defined and is not included in the 2022 WHO list of hematological neoplasms nor in the recent International Consensus Classification. Moreover, other LPDs exhibit an overlapping expression of some antigens present on CLL cells. Furthermore, the biological relationship between phenotypically typical CLL, CLL with a minor alteration, and non-CLL leukemic LPD with some CLL-like findings is unknown, and a definitive diagnosis of aCLL based on laboratory and clinical data can be difficult to make. In the future, molecular studies will play a key role in characterizing aCLL and related diseases. Today, a diagnosis of aCLL has no therapeutic implications, as it requires a similar treatment to classic CLL. However, following the recent introduction of targeted drugs designed on the basis of biological mechanisms, a better definition of aCLL is needed to provide the optimal treatment for each group of patients. 

## Figures and Tables

**Figure 1 cancers-15-04427-f001:**
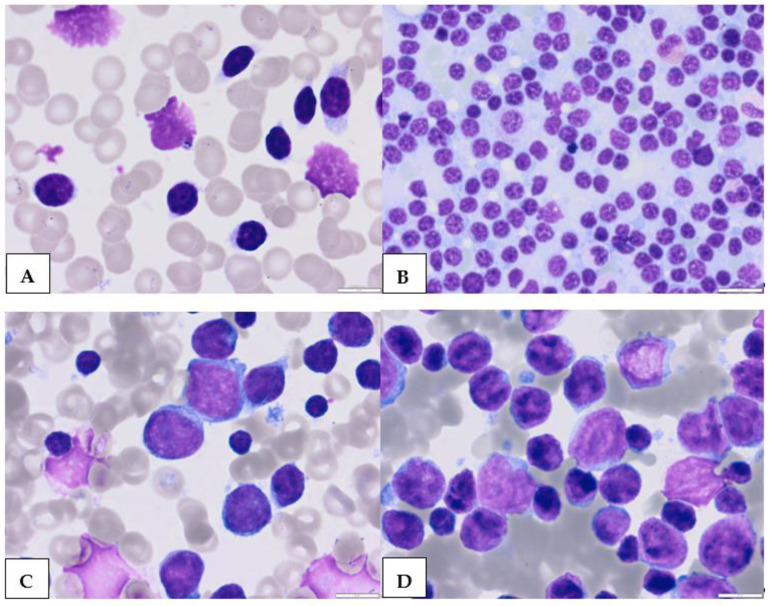
Morphological features of classic (**A**,**B**) and large (**C**,**D**) CLL cells. Mature CLL cells are lymphocytes with a narrow border of cytoplasm and partially aggregated chromatin in a dense nucleus ((**A**)—peripheral blood, (**B**)—bone marrow). Large atypical CLL cells ((**C**)—peripheral blood, (**D**)—bone marrow) (magnification 63×).

**Figure 2 cancers-15-04427-f002:**
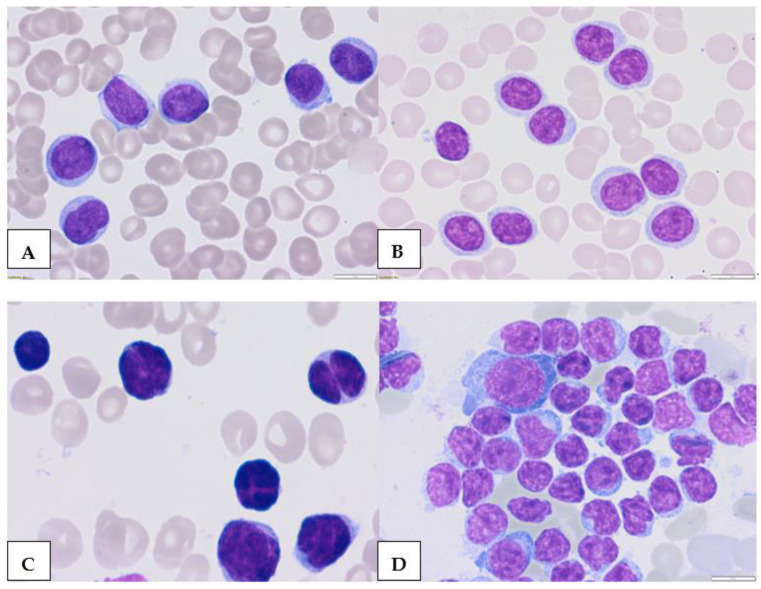
Morphologic features of atypical CLL cells. Prolymphocytes in peripheral blood (**A**,**B**) and cleaved cells (**C**)—in peripheral blood, (**D**)—bone marrow (magnification 63×).

**Figure 3 cancers-15-04427-f003:**
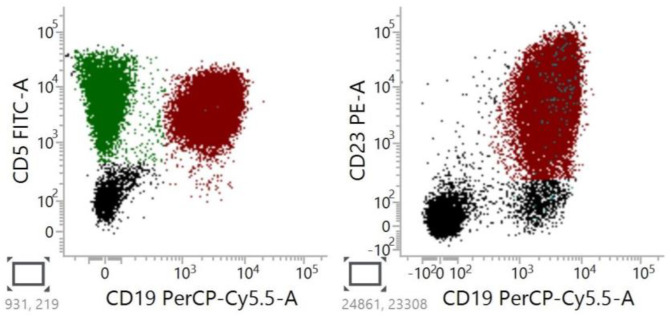
Representative flow cytometry of classic CLL cells showed positive expressions of CD5, CD19, and CD23. The population of CD5+/CD19+ and CD23+/CD19+ cells is marked with red color.

**Figure 4 cancers-15-04427-f004:**
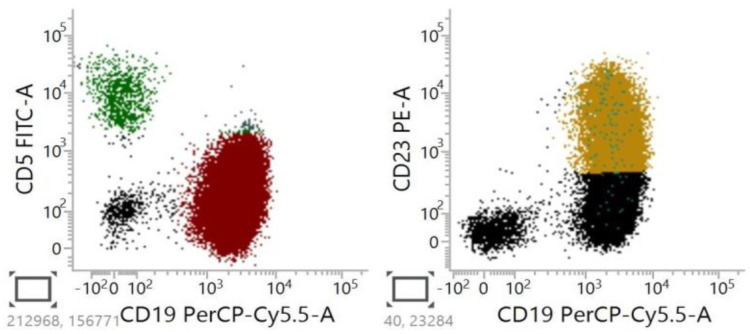
Representative flow cytometry of atypical, CD5-negative CLL cells showed positive expression of CD19, CD23, and no expression of CD5. Red color indicates the CD5−/CD19+ cell population, yellow color indicates the CD23+/CD19+ population.

**Figure 5 cancers-15-04427-f005:**
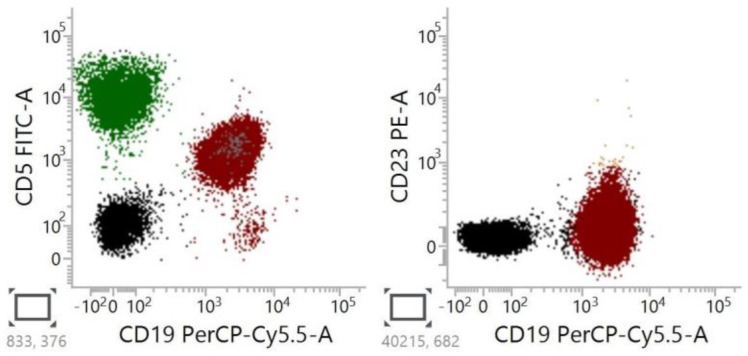
Representative flow cytometry of CLL patients showed positive expression of CD5 and CD19, and negative expression of CD23. The population of CD5+/CD19+ and CD23−/CD19+ cells is marked with red color.

**Table 1 cancers-15-04427-t001:** Larger studies comparing patients with CD5-negative and CD5-positive CLL.

Authors/Reference	No of pts CD5− vs. CD5+	Median Age: CD5− vs. CD5+	Definition of CD5− CLL	PB CD5− vs. CD5+	Treatment CD5− vs. CD5+	Survival CD5− vs. CD5+	Commentary
Cartron et al., 1998 [41]	42 vs. 79	68/64.8	<5% of mononuclear cells	Hb: 126/137 G/L (*p* = ns) PLT: 216/200 × 10^9^/L (*p* = ns) Lymphocytes: 27.3/32.7 × 10^9^/L × 10^9^/L (*p* = ns)	No initially treated:64.3% vs. 29.1%	52% at 120 m vs. 6% at 90 m (*p* = 0.97)	CD5− CLL expressed a higher level of surface of immunoglobulin and had more frequently isolated splenomegaly.
Efstathiou et al., 2002 [42]	29 vs. 29	68.8/68.4	<5% of mononuclear cells	Hb: 131/10.5 G/L (*p* = ns) (*p* < 0.05) PLT: 211/198 × 10^9^/L (*p* = ns) Lymphocytes:38.2/39.6 × 10^9^/L (*p* = ns)	No initially treated:72.4% vs. 24.1%	Median:97.2 m vs. 84.0 m (*p* = 0.0025)	Splenomegaly, lymph node involvement, and hemolytic anemia less common in CD5− CLL. CD5− CLL patients had a more favorable prognosis compared with CD5+ patients
Demir et al., 2017 [43]	19 vs. 105	65.8/66.5	<20% of mononuclear cells	HB: 133/127 g/L (*p* = 0.180) PLT: 144/160 × 6 × 10^9^/L (*p* = 0.044) Neutrophils: 3.5/3.36 × 10^9^/L (*p* = 0.169) Lymphocytes: 43.2/36.7 × 10^9^/L (*p* = 0.001).	NR	84.2% vs. 90.5% at 5 yr (*p* = 0.393)	Lymphadenopathy less frequent in CD5− (*p* = 0.029). Splenomegaly more frequent in CD5− (*p* = 0.029). No difference in clinical stage, autoimmune phenomena, hemo globin and neutrophil count, and survival
Kurec et al., 1992 [45]	12 vs. 27	66/67	<20% of lymphoid cells	Hb: 11.2/13.7 g/L PLT: 172/175 × 10^9^/L WBC: 88 × 10^9^/L/60 × 10^9^/L	NR	55% vs. 90% at 5 yr	Lack of CD5 antigen was with more advanced stage of disease and poor patient survival.

Abbreviations: CLL—chronic lymphocytic leukemia; Hb—hemoglobin; ND—not differ; NR—not reported; PLT—platelets; WBC—white blood cells.

**Table 2 cancers-15-04427-t002:** Immunophenotypic differential diagnosis of typical CLL, atypical CLL, and MCL.

Disease	Markers											
	CD19 CD20 CD22	CD5	CD23	FMC7	CD200	CD45	CD43	CD180	Cyclin D1	CD79b	LEF1	SOX11
Typical CLL	+ (dim)	+	+ (strong)	−	+ (strong)	−/+	+	+/− (week)	−	+/− (week)	+	−
Atypical CLL	+	−	−	−	+	+	+	+	−	NR	+/−	NR
MCL	+ (strong)	−	−	+/−	−	+	−/+	+ (week)	+	+/−	−	+
References	7,8 36–40, 69	18,29, 36–40	18,29, 36–40	64,73	18,29, 36–40, 68–72	74,75	64,67, 76–79	64,76, 84–86	52,68	64,68	68,73, 81,82	68, 87–89

Abbreviations: CLL—chronic lymphocytic leukemia; MCL—mantle cell lymphoma; +—positive; +/−—mostly positive; −/+—mostly negative; NR—not reported.

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
