# Peer review of "Atypical Chronic Lymphocytic Leukemia—The Current Status"

_cancers, 2023, doi:10.3390/cancers15184427_

Round 1

Reviewer 1 Report

Abstract

The definition of prolymphocyte in the abstract is wrong.

The sentence “However, atypical CLL is WHO 73 HAEM5 nor the ICC [7,8]” makes no sense to me. Do the authors mean that aCLL is not included in HAEM5 not in the ICC? Do they mean that they are reviewing an entity that is not included in the classifications? The review should centered on prolymphocytic progression of CLL or on accelerated CLL .

In line 87-89 the authors state “This subtype of CLL is commonly associated with trisomy of chromosome 12, a higher intensity of cell B markers, and a poor prognosis”. A paper should be quoted. If the citation refers to a pre chemoimmunotherapy era the sentence is no longer correct.

“High intensity of B-cell markers” I guess it ref er to the sIg expression. Please specify

Author Response

Reviewer 2.

The authors reviewed the current status of atypical CLL. The review is clear, scientifically accurate and well-written. 

Response: We thank the Reviewer for positive review of our paper

I have the following minor concerns:

Introduction: The authors state: However, atypical CLL is WHO HAEM5 nor the ICC [7,8]. Please correct the phrase, because it does not make sense. the authors obviously mean that aCLL in not included in both cclassifications (WHO and ICC). 

Response:   We corrected the sentence (However, atypical CLL is not included in the WHO HAEM5 nor the ICC classification  [7,8].

  1. Atypical immunophenotype: The authors state: '....becomes more challenging when CD23 or CD5 is not expressed by the leukemic cells. Please add the sentence that usually in clinical practice in such cases lacking CD5 or CD23, the discussion starts whether  it is an aCLL or not. 

Response: We dded the sentence: In clinical practice, in cases lacking CD5 or CD23, the diagnostic process begins with confirming, or rejecting aCLL.

  1. Will immunophenotype or genetic/molecular analysis make morphology sound much less important (if not at all important) for diagnosing aCLL in the future? Please answer that to a novel paragraph and discuss. 

Response:The novel paragraph is added: In the recent years morphology sound is much less important than immunophenotype or genetic/molecular analysis for diagnosis of aCLL and probably will be even less important in the future. According to the recent recommedations assessment of the immunophenotype plays a crucial role in the diagnosis of CLL and the morphology fof CLL cells is less important in the diagnostic procedure. A significant diagnostic challenge is CLL without CD5 or CD23 antigens expression, defined as aCLL and these subtypes of CLL need further laboratory and clinical investigation. In addition, cytogenetic and molecular findings, especially TP53 ond IGHV mutational status provide important diagnostic and prognostic information that influence treatment decisions, even in the era of targeted drugs. Identification of less common genetic abnormalities should deliver better characteristics of aCLL, and influence treatment selection in the future.

  1. Please make an additional table summarizing the knowledge described in part 6 of the manuscript (differential diagnosis of aCLL), especially regarding expression of CD200, CD45 and CD43. The table must also contain the references and the immunophenitypic findings, along with comments for the differential diagnosis of aCLL.

Response: The novel table (No 2) is added

  1. Are there published series of aCLL and what is the OS and the overall prognosis? Which are the largest series? Please discuss in the paper. 

Response: We have added  several new sentences on this topic in the sub-chapter 3.  The CD5-negative patients frequently expressed a higher level of surface immunoglobulin and were more likely to present with splenomegaly [42].

 The  largest study, reported by Cartron et al [42], compared 42 consecutive patients with CD5- CLL, observed from 1985 to 1991, with 79 with CD5+ CLL. . However, this study was performed before the immunochemotherapy era and the patients were treated either with chlorambucil or with CHOP (cyclophosphamide, doxorubicine, vincristine and prednisone). At the time of data analysis, the median survival time was not reached and no significant difference was observed between groups: overall survival was 52% at 120 months in the CD5- group, and 66% at 90 months in the CD5+ group (p = 0.97). 

In another study, CD5-negative patients were found to exhibit milder disease symptoms and longer survival than the CD5-positive patients [43]. Efstathiou et al. compared the clinical and biological characteristics of 29 CD5-negative CLL patients with a control group of 29 sex- and age-matched, consecutive CD5-positive CLL patients [43]. In the CD5-negative group, splenomegaly, lymph node involvement and hemolytic anemia were significantly less common than in the  CD5-positive group. In this study, CD5- patients had a more favorable prognosis than the CD5+ CLL patients [43]. The patients in both groups were treated  with chlorambucil and prednisone, fludarabine or COP (cyclophosphamide, vincristine and prednisone). The CD5- patients demonstrated significantly longer median OS (97.2 months) than the CD5+ patients (84.0 months, p = 0.0025).  

In a more recent study, Demir et al compared 19 consecutive CD5-negative CLL cases observed from 2009 to 2015 with 105 CD5-positive CLL patients [44]. Lymphadenopathy was less common (31.5%) of the CD5-negative group than the CD5-positive group (51.4%; p=0.029) but splenomegaly was more common in the CD5-negative group (42.1%) than the CD5-positive group (16.1%, p=0.029). However, no difference in Binet staging or median neutrophil count was noted between the groups. The CD5-negative patients also exhibited a higher mean lymphocyte count (43.2×109/L) than the CD5-positive patients (36.7×109/L, p=0.001). The five-year survival rate was 84.2% in CD5-negative patients and 90.5% in the CD5-positive patients (p=0.393). However, no treatment details are available.

Romano et al. analyzed a cohort of 400 CLL patients, including 13 with a CD5-negative phenotype. No significant differences in clinical course and survival were observed between the CD5-positive and CD5-negative CLL cells [47]. Elsewhere, Kurec et al. found that CD5-negative CLL patients exhibited a lower hemoglobin level and higher disease stage (Rai’s classification) at diagnosis; they also demonstrated worse OS, with a five-year survival rate of 55% in CD5-negative patients and >90% in CD5-positive patients [46]. 

  1. The authors state: 'Patients with CLL-like MCL have distinctive immunophenotypic features that are useful for distinguishing MCL from CLL'. Who and what are these distinctive immunophenotypic features? Please explain, describe these unique features and discuss. 

Response: We included this  point in  chapter 6: In the majority of patients, CLL can be differentiated from MCL and other lymphoid malignancies by flow cytometry (Table 2)[68]. In particular, patients with CLL-like MCL are more likely to display moderate to high expression of B-cell antigens and Sig light chain. In addition, while CLL cells mostly present dim to negative expression of B-cell antigens such as CD20, CD22, and CD79b, dim expression is uncommon in CLL-like MCL. In addition, MCL cases are positive for cyclin D1 and SOX11, and negative for CD10, CD23, CD200, and LEF1 [68]. However, differential diagnosis may be challenging if CD23 is not expressed by the CLL lymphocytes, or where CD23 is expressed by MCL, as the two entities share many similarities [69]. In such cases, MCL diagnosis should be confirmed by immunohistochemical cyclin D1 detection, together with other cytofluorimetric readings, as well as cytogenetic or molecular testing

Another antigen that may be useful in the differentiation of CLL from other lymphoproliferative disorders is CD43 [76-78]. CD43 is a transmembrane sialogclyoprotein expressed on a subset of B-cells, T-cells, monocytes and granulocytes, but not on follicular lymphoma or MCL. It is expressed in both aCLL and typical CLL [79]. Moreover, the positive rate of CD43 was higher in CLL than in MCL  A recent study showed that CD43 and CD200 can differentiate CLL from non-CLL LPD with higher accuracy than the Matutes scoring system [64,75].   In addition, no difference in CD43 and CD200 expression has been observed between classic CLL and aCLL patients. In other study, CD81, CD5, CD23 and CD200 were found to be useful markers for distinguishing CLL from other LPDs [80].   

Lymphoid enhancer-binding factor (LEF1) staining was identified in up to 95% of patients with CLL [81,82]. Lower LEF1 expression was found in CLL patients with atypical immunophenotypic or morphological findings compared to classic CLL [81].  In contrast, LEF1 expression is only observed in 4 to 9% of MCL patients [81,82].     

CD180 is a toll-like receptor homolog protein expressed on B cells [83]. Although several studies have confirmed its expression on indolent lymphoid malignancies, CD180 is weakly expressed in CLL and MCL compared with normal B cells [84,85]. However, CD180 has been suggested as a marker for aCLL [86]. The presence of double-positive CD43 and CD180, together with other B-cell surface markers, can improve the sensitivity of the diagnosis in CLL patients negative for CD5 or CD23 staining [64].  Despite this, it can be difficult to make a definitive diagnosis of aCLL based on laboratory and clinical data alone. Other markers for the differential diagnosis of CLL are CD79b and FMC7. A recent study found that CD79b and FMC7 tend to express negatively expression in CLL and positively in MCL [64].

Finally, SOX11 and t(11;14) are important markers for differentiating CLL from MCL[87-89], with SOX11 being consistently detected in MCL but not in CLL [89]. However, no data is available on SOX11 expression in aCLL.

Reviewer 2 Report

The authors reviewed the current status of atypical CLL. The review is clear, scientifically accurate and well-written. 

I have the following minor concerns:

1. Introduction: The authors state: However, atypical CLL is WHO HAEM5 nor the ICC [7,8]. Please correct the phrase, because it does not make sense. the authors obviously mean that aCLL in not included in both cclassifications (WHO and ICC). 

2. Atypical immunophenotype: The authors state: '....becomes more challenging when CD23 or CD5 is not expressed by the leukemic cells. Please add the sentence that usually in clinical practice in such cases lacking CD5 or CD23, the discussion starts whether  it is an aCLL or not. 

3. Will immunophenotype or genetic/molecular analysis make morphology sound much less important (if not at all important) for diagnosing aCLL in the future? Please answer that to a novel paragraph and discuss. 

4. Please make an additional table summarizing the knowledge described in part 6 of the manuscript (differential diagnosis of aCLL), especially regarding expression of CD200, CD45 and CD43. The table must also contain the references and the immunopenitypic findings, along with comments for the differential diagnosis of aCLL. 

5. Are there published series of aCLL and what is the OS and the overall prognosis? Which are the largest series? Please discuss in the paper. 

6. The authors state: 'Patients with CLL-like MCL have distinctive immunophenotypic features that are useful for distinguishing MCL from CLL'. Who and what are these distinctive immunophenotypic features? Please explain, describe these unique features and discuss. 

Author Response

Reviewer 3.

Comments and Suggestions for Authors

The work is well written and provides many examples of literature to support and deepen the topic covered.

Comparisons to clarify the differential diagnosis with other pathologies are also extensive and well articulated. Definitely a nice article that explores a little-known but very useful topic in everyday clinical practice. I have nothing more to say and I recommend the publication.

Response: We thank the Reviewer for this comment.

 Needed a rearrangement of the Layout in some places.

Response:.The rearrangement of the Layout in some places was done.

At the beginning of line 114 there is an error and in line 232, point 3.2 must be brought to the head of the line and put in bold.

Response:.Corrected as indicated

Comments on the Quality of English Language: NO concern on the quality of the english

Reviewer 3 Report

The work is well written and provides many examples of literature to support and deepen the topic covered.

Comparisons to clarify the differential diagnosis with other pathologies are also extensive and well articulated. Definitely a nice article that explores a little-known but very useful topic in everyday clinical practice. I have nothing more to say and I recommend the publication. Needed a rearrangement of the Layout in some places.

At the beginning of line 114 there is an error and in line 232, point 3.2 must be brought to the head of the line and put in bold.

NO concern on the quality of the english

Author Response

(The authors gave the same response as above.)

Reviewer 4 Report

The review article “Atypical chronic lymphocytic leukemia – the current status” summarized the difference between CLL and aCLL concerning morphological findings, phenotype, genotype, and prognosis. Additionally, the difference between MCL and aCLL was described well. This review article was very informative for a lot of readers. However, the volume of manuscript was too much for readers to understand. Therefore, I considered that there were several comments for the purpose to improve your review article.

1.       Valuable tables can contribute to understand well for readers in review article. I considered that the difference among CLL, MCL, and aCLL was very complicated for differential diagnosis. Therefore, the author should add a table for differential diagnosis among them.

2.       Table 1 showed the clinical prognosis between CD5 negative and positive CLL. Was there the difference of treatment between CD5 negative and positive CLL in these four large studies? 

3.       Do you have any data about clinical symptoms between CLL and aCLL?

Author Response

Reviewer 4.

Comments and Suggestions for Authors

The review article “Atypical chronic lymphocytic leukemia – the current status” summarized the difference between CLL and aCLL concerning morphological findings, phenotype, genotype, and prognosis. Additionally, the difference between MCL and aCLL was described well. This review article was very informative for a lot of readers. However, the volume of manuscript was too much for readers to understand. Therefore, I considered that there were several comments for the purpose to improve your review article.

Response: We thank the Reviewer for these comments .  

Valuable tables can contribute to understand well for readers in review article. I considered that the difference among CLL, MCL, and aCLL was very complicated for differential diagnosis. Therefore, the author should add a table for differential diagnosis among them.

Response: The table 2 is added according to the Reviewer suggestion. In addition, additional text and references in the subchapter 5 are added.

In the majority of patients, CLL can be differentiated from MCL and other lymphoid malignancies by flow cytometry (Table 2)[68]. In particular, patients with CLL-like MCL are more likely to display moderate to high expression of B-cell antigens and Sig light chain. In addition, while CLL cells mostly present dim to negative expression of B-cell antigens such as CD20, CD22, and CD79b, dim expression is uncommon in CLL-like MCL. In addition, MCL cases are positive for cyclin D1 and SOX11, and negative for CD10, CD23, CD200, and LEF1 [68]. However, differential diagnosis may be challenging if CD23 is not expressed by the CLL lymphocytes, or where CD23 is expressed by MCL, as the two entities share many similarities [69]. In such cases, MCL diagnosis should be confirmed by immunohistochemical cyclin D1 detection, together with other cytofluorimetric readings, as well as cytogenetic or molecular testing

  1. Table 1 showed the clinical prognosis between CD5 negative and positive CLL. Was there the difference of treatment between CD5 negative and positive CLL in these four large studies? 

Response: .No, the treatment was the same in both arms

  1. Do you have any data about clinical symptoms between CLL and aCLL?

Response:. The available data from some studies are mentioned in the text and marked yellow.

Round 2

Reviewer 1 Report

The authors addressed the criticisms

Minimal revision of te language